# Mass Spectrometry for the Monitoring of Lipoprotein Oxidations by Myeloperoxidase in Cardiovascular Diseases

**DOI:** 10.3390/molecules26175264

**Published:** 2021-08-30

**Authors:** Catherine Coremans, Cédric Delporte, Frédéric Cotton, Phillipe Van De Borne, Karim Zouaoui Boudjeltia, Pierre Van Antwerpen

**Affiliations:** 1RD3-Pharmacognosy, Bioanalysis and Drug Discovery, Faculty of Pharmacy, Université Libre de Bruxelles, 1050 Brussels, Belgium; cedric.delporte@ulb.be (C.D.); pierre.van.antwerpen@ulb.be (P.V.A.); 2Laboratoire Hospitalier Universitaire de Bruxelles (LHUB-ULB), Department of Clinical Chemistry, Université Libre de Bruxelles (ULB), 1000 Brussels, Belgium; frederic.cotton@lhub-ulb.be; 3Department of Cardiology Erasme Hospital, Université Libre de Bruxelles, 1070 Brussels, Belgium; philippe.van.de.borne@ulb.be; 4Laboratory of Experimental Medicine (ULB 222 Unit), CHU-Charleroi, ISPPC Hôpital Vésale, Université Libre de Bruxelles, 6110 Montigny-Le-Tilleul, Belgium; karim.zouaoui.boudjeltia@ulb.be

**Keywords:** apolipoproteins, oxidation, peptide quantification, LC-MS/MS, lipoprotein quality

## Abstract

Oxidative modifications of HDLs and LDLs by myeloperoxidase (MPO) are regularly mentioned in the context of atherosclerosis. The enzyme adsorbs on protein moieties and locally produces oxidizing agents to modify specific residues on apolipoproteins A-1 and B-100. Oxidation of lipoproteins by MPO (Mox) leads to dysfunctional Mox-HDLs associated with cholesterol-efflux deficiency, and Mox-LDLs that are no more recognized by the LDL receptor and become proinflammatory. Several modification sites on apoA-1 and B-100 that are specific to MPO activity are described in the literature, which seem relevant in patients with cardiovascular risk. The most appropriate analytical method to assess these modifications is based on liquid chromatography coupled with tandem mass spectrometry (LC-MS/MS). It enables the oxidized forms of apoA-1and apoB-100 to be quantified in serum, in parallel to a quantification of these apolipoproteins. Current standard methods to quantify apolipoproteins are based on immunoassays that are well standardized with good analytical performances despite the cost and the heterogeneity of the commercialized kits. Mass spectrometry can provide simultaneous measurements of quantity and quality of apolipoproteins, while being antibody-independent and directly detecting peptides carrying modifications for Mox-HDLs and Mox-LDLs. Therefore, mass spectrometry is a potential and reliable alternative for apolipoprotein quantitation.

## 1. The Oxidative Activity of Myeloperoxidase on Lipoproteins in the Context of Atherosclerosis

Atherosclerosis is a chronic inflammatory disease which begins with a deficiency in the endothelium function from the artery wall through a set of dysfunctions including inflammation and oxidative stress [1,2]. The synthesis of inflammatory mediators initiates the oxidation of LDLs around the vascular wall, which induces the production of inflammatory factors [3]. The oxidation of LDLs also promotes the development of foam cells by accumulation of LDL-derived cholesterol in macrophages and leads to the formation of atheromatous plaque [4].

The myeloperoxidase-nicotinamide adenine dinucleotide phosphate oxidase system (MPO—Nox or NADPH oxidase) is a major source of LDL-oxidizing agents in the vascular wall, and the most relevant. The NADPH oxidases are complex enzymes with several subunits, depending on their location. These enzymes produce superoxide anion (O_2_^−^) in the phagosome lumen or extracellular fluids, from an oxygen molecule using NADPH. Superoxide anion undergoes dismutation, spontaneously or via superoxide dismutase (SOD) [5], and gives hydrogen peroxide (H_2_O_2_) [6]. Smooth muscle cells contain Nox 1 and Nox 4, whereas macrophages express mainly Nox 2 [7] (cfr Figure 1). As Nox 2 is strongly expressed within macrophages, it is consequently abundant in plaque of atherosclerotic arteries [7]. In endothelial cells, Nox 2 and Nox 4 are mainly expressed [8]. Activation of the endothelial Nox 2 alone is sufficient for the initiation of atherosclerosis and is an indispensable contributor for the progression of atherosclerosis in other types of cells [9]. An atherogenic level of LDL leads to an increase in oxidative species by stimulating Nox in the endothelium [10]. Nox enzymes can be induced under atherogenic conditions such as Nox4 in monocytes and macrophages. It mediates oxidized low density lipoprotein-induced macrophage death [11].

Therefore, MPO has a key role in the oxidation of LDLs by using H_2_O_2_ generated through activation of Nox and SOD. In physiologic conditions, MPO is mostly present in azurophilic granules of neutrophils and a little in monocytes [12]. MPO produces strong oxidative species that struggle infections by consuming H_2_O_2_ and (pseudo)halide anions to form oxidants (cfr Figure 1). It binds to the bacterial membrane and produces locally mainly hypochlorous acid (HOCl) to oxidize macromolecules from the wall of ingested bacteria, such as proteins, unsaturated lipids and even nucleic acids [13]. MPO activity includes two cycles: the halogenating and the peroxidase cycles. The halogenation cycle is referred as the MPO/H_2_O_2_/X^−^ system, where native MPO reacts with H_2_O_2_ producing Compound I and returning to its native form by oxidizing a (pseudo-)halide. The latter are mainly chloride anions (X^−^such as Cl^−^) but also bromide, iodide and thiocyanate anions, and they lead to production of the corresponding hypo(pseudo-)halogenous acids. Hypochlorous acid (HOCl) is mainly produced, since chloride anions are the most abundant in vivo halides, even with a less favorable kinetic constant of oxidation than bromide, for example. 

In the atherosclerosis context, MPO is important due to its association with coronary artery diseases and is considered a circulating marker of cardiovascular diseases (CVD) [14]. Elevated levels of circulating MPO are observed in patients with coronary artery diseases, unstable angina, and acute myocardial infarction [14]. Its concentration at plasma level has been reported to be higher in myocardial infarction (MI) patients (55 ng/mL) as compared to control subjects (39 ng/mL) [15]. As the measurement of its activity in biological samples has emerged essential for clinical investigations, immunological methods to measure active and total myeloperoxidase have been developed [16].

The modifications of the protein moiety via the MPO/H_2_O_2_/Cl^−^ system have deleterious effects in the context of atherosclerosis. As MPO is also present in extracellular medium [17], the oxidation of LDLs by MPO in Mox-LDLs can occur at the surface of endothelial cells, and would not be limited to the subendothelial space [18]. Endothelial cells, which are stimulated by angiotensin II and LDLs, are indeed capable of generating H_2_O_2_ thanks to Nox 2 [19]. Furthermore, reactivity of amino acids towards HOCl have been intensively studied in proteins. HOCl reacts first with sulfur amino acids, such as methionine (Met). Methionine is an accessible amino acid and highly oxidizable in sulfoxide methionine (oxMet) [13]. The ability of sulfur compounds to inhibit MPO is frequently suggested and Met is suspected to act as a “last chance” antioxidant for proteins [20]. Some residues are considered hallmarks of inflammatory tissue injury and have been detected during inflammation. Tryptophan (Trp) is very sensitive to oxidation and can be targeted by MPO around the benzene moiety of indole (5-mono or 5,7-dihydroxytryptophan) and the pyrrole moiety (2-oxo-tryptophan in equilibrium with 2-hydroxy-tryptophan). *N*-formylkynurenine and kynurenine forms can also be produced [21]. Tyrosine (Tyr) can be modified on the phenol core in chlorinated derivatives (3-chlorotyrosine (Cl-Tyr), 3,5-dichlorotyrosine) and nitrated derivatives (3-NO_2_Tyr) [22,23].

Cl-Tyr has been identified as a signature residue for MPO-activity, strongly present in lipoproteins in human atherosclerotic lesions [22]. The abundance of Cl-Tyr has been highlighted in LDLs isolated from human atherosclerotic wall, and MPO is the only known human enzyme able to produce hypochlorous acid (HOCl) that chlorinates Tyr. This phenomenon is also observable in high-density lipoproteins (HDL) isolated from blood of humans with established coronary artery disease [24,25]. The possibility that MPO may promote atherogenesis by inhibiting the antiatherogenic effects of HDL and inducing oxidized-LDL, should encourage to keep investigating.

Lipoproteins oxidation impairs the cholesterol transport. One of the key cholesterol carriers in the body is LDLs with apoB-100, which allows the recognition of LDLs by the LDL-receptor (LDLR) [26]. In the reverse cholesterol transport mediated by HDLs, apoA-1 is the key in the interaction between HDLs and the HDL-receptor (HDLR), the lecithin–cholesterol acyltransferase and the transporter ATP-binding cassette A1 (ABCA1) [27]. MPO-modifications of apoA-1 produce Mox-HDLs that are dysfunctional HDLs with oxidation of Trp and Met residues or nitration and chlorination of Tyr residues. Two mean different residues are currently considered as specific biomarkers of atherogenic activity of MPO on apoA-1: tyrosine 192 (Tyr 192) and tryptophan 72 (Trp 72) [22,28,29]. Some other MPO-modified residues have been identified (cfr Table 1), but these two seem the most relevant.

In order to study the impact of Mox-HDLs and more precisely oxidized-Trp 72 (oxTrp 72), a high affinity monoclonal antibody (mAb) recognizing apoA-1 modified by the MPO/H_2_O_2_/Cl^-^ system has been developed [35]. The oxindolyl alanine residue (2-OH-Trp) corresponding to the Trp 72 of apoA-1 was identified as the immunogenic epitope. The circulating apoA-1 presents few oxTrp 72, while 20% of apoA-1 of the atherosclerotic plaque contain oxTrp 72. This modified-apoA-1 is characterized by a low lipid content, a loss of cholesterol acceptor activity, and a pro-inflammatory activity on endothelial cells [35]. The biogenesis of HDL is also impaired. A high level of oxTrp 72-apoA-1 has been associated with an increased risk of CVD. Then, circulating oxTrp 72-apoA1 could be used as monitor of pro-atherogenic process in the artery wall [35].

Interactions have been investigated between MPO and native/modified LDLs (cfr Table 2) [36]. The identification of MPO-mediated modifications apoB-100 in vitro have been confirmed with (Mox-)LDL isolated from patients at high cardiovascular (CV) risk. Among the MPO-targeted residues and specific to patients, oxMet 2499, oxTrp 2894/3606, Cl-Tyr 76/102/125/749, and dioxTrp 4369 have been identified. A monoclonal antibody against Mox-LDLs have been developed, characterized as AG-948F4A2 (AG9) and targeting only apoB modifications by MPO. However, the exact structure of the recognized epitope has not been yet identified [37,38].

Oxidized residues of apo-A1 and apoB-100 are present in patients with CV risks or diseases, and their clinical relevance regarding their deleterious effects in vivo should be investigated. Moreover, MPO plasma concentration is correlated to CV risk and its activity is illustrated by oxidation, chlorination and nitration of lipoproteins. Therefore, the evaluation of causal factors of atherosclerosis, such as MPO-oxidized apoB-100 and apoA-1, could bring additional information to improve the assessment of CV risk, in addition to measurement of traditional risk factors.

## 2. Reconsideration of the Actual Cardiovascular Risk Assessments

Considering the deleterious effects of MPO-oxidized lipoproteins, a more accurate evaluation of the absolute risk for a first CV event is needed for treatment recommendations to provide individual and appropriate cure. The first step is to identify classical risk factors by determining traditional risk factors for CVD, such as hypertension, smoking, diabetes, premature family history of CVD, kidney disease, obesity and dyslipidaemia [45]. The Framingham Heart Study (FHS) [46] has highlighted the need for considering all risk factors, by demonstrating that a high rate of cholesterol was insufficient to explain CV events. Lipid profile is strongly recommended in the assessment of the CV risk and screening for dyslipidaemia is always indicated in subjects with clinical manifestations of CVD. A first measure of total cholesterol is used in *Systematic Coronary Risk Evaluation* (SCORE) system [47]. The SCORE table is divided in two charts, the low-risk ones and the high-risk ones, depending on the country and whether their national cardiology societies belong to the European Society of Cardiology (ESC) or not [48]. Identification of dyslipidemia through LDL cholesterol, triglycerides, HDL cholesterol and non-HDL cholesterol measurements are recommended for screening, risk assessment, diagnosis and treatment monitoring. Other parameters are considered, such as apoB, Lp(a) and the ratio apoB/apoA-1 or non-HDL-C/HDL-C, in specific cases such as a high level of triglycerides.

The current guidelines for the management of dyslipidaemias aim at lowering the lipid blood content to reduce cardiovascular risk. ESC and European Atherosclerosis Society (EAS) regularly update guidelines to summarize available evidence and recommendations in management of an individual patient to reduce atherosclerotic CV risk in adults [49].

LDL-C is the primary target for diagnosis and treatment in CV risk management. Numerous studies have consistently demonstrated a log–linear relationship between plasmatic LDL-C and the risk of atherosclerotic CV disease (ASCVD) [50]. There is strong evidence that LDL-C is causally associated with the risk of ASCVD, and that lowering LDL-C reduces the risk of ASCVD proportionally to LDL-C reduction [51]. As circulating LDL particles are also estimated by apoB, the reduction in LDL-C is mirrored by a reduction in cholesterol carried by these particles [51]. Therefore, lowering LDL-C by reducing LDL particle mass is proportional to the absolute reduction in LDL-C, as reduction in LDL-C and LDL particles are concordant [50]. In contrast, lowering LDL-C by drastically modifying their composition is proportional to the absolute change in LDL particle concentration as measured by a reduction in apoB [51].

There is a consistent inverse association between HDL-C and risk of ASCVD [52]. In contrast, there is no evidence that HDL-C is causally associated with the risk of ASCVD, or that therapeutically increasing plasma HDL-C reduces the risk of CV events [53,54]. However, it must be interpreted carefully as most genetic variants of HDL-C are also associated with changes in TGs and/or LDL-C.

### 2.1. Actual Measurements

Current measurements of lipid profile are essentially performed with ready-to-use kits. Total cholesterol assay kit uses a simple method to quantify total cholesterol, free cholesterol, and cholesterol esters in serum and plasma samples. As a reminder, cholesterol is made up of LDL-cholesterol, HDL-cholesterol, and VLDL-cholesterol. The classical assay to measure total cholesterol is an enzymatic reaction coupled to a colorimetric test by incubating with a mixture of enzymes (cholesterol esterase, cholesterol oxidase and peroxidase), detergents, 4-aminoantipyrine and buffer [55]. Cholesterol oxidase reacts with free cholesterol to produce cholest-4-en-3-one and hydrogen peroxide. The latter reacts, in turn, with a probe to generate color at a specific wavelength (570 nm) and fluorescence (Ex/Em = 538/587 nm) (“Cholesterol Assay Kit—HDL and LDL/VLDL (Ab65390) | Abcam” 2021). When total cholesterol is measured, cholesterol esterase is used to hydrolyze cholesteryl ester into free cholesterol and fatty acid. The amount of cholesterol ester is calculated by subtracting free cholesterol from total cholesterol. To only assess HDLs or LDLs/VLDLs, it is possible to separate these lipoproteins by specific precipitation. HDL-C is measured in plasma by analyzing the amount of cholesterol associated with these particles. Lipoproteins (chylomicrons, VLDL and LDL) are precipitated by the addition of phosphotungstic acid and magnesium chloride. After centrifugation, the clear supernatant contains the HDL fraction which is tested with the cholesterol colorimetric test [55]. The same principle is used for LDL-C, with a first incubation of the sample which aims at masking VLDL and chylomicrons by a first reagent (MgSO_4_, α-cyclodextrin sulfate and dextran sulfate). The second incubation use a specific buffer which allows selective solubilization of LDL lipoproteins. In general, LDL-C is calculated with the Friedwald equation, although its resultats are contested [56].

The determination of LDL-cholesterol by Friedewald’s formula is as follows: LDL-cholesterol = total cholesterol − HDL-cholesterol − (triglycerides/5). This cannot be used in the presence of abundant chylomicrons or intermediate density lipoproteins (for example, during post-meal period), or when triglyceridemia exceeds 4 g/L. Methods for determining LDL-cholesterol have been compared to the calculated LDL-cholesterol. In patients with normal triglyceridemia values, the correlation between the two methods is satisfying. However, there is a significant difference between calculated and measured LDL cholesterol values. In addition, there is an overestimation of the calculated LDL-cholesterol compared to the measured LDL-cholesterol in 17% of cases [57].

With biotechnological progress, immunoassays have been developed to quantify analytes such as lipoproteins and apolipoproteins, through the formation of a stable complex between the analyte and a specific antibody. The detection of this complex is ensured by labelling of the analyte, the antibody, or another compound of the assay. Immunoturbidimetry test is a photometric measurement of the turbidity induced by the immune-complex precipitates. The most common immunoassay is the enzyme-linked immunosorbent assay (ELISA). The principle of the ELISA assay is to detect apolipoproteins indirectly via antibodies recognizing specific epitopes on the proteins. There are four types of ELISA method: direct, indirect, sandwich and competitive. The indirect ELISA is the most used. The antibody to be analyzed from serum will bind to its antigen in wells. Then, the detection antibody will interact in turn with the endogenous antibody. The detection antibody is coupled to an enzyme, which transforms its substrate into a colorful and measurable product. The intensity of the coloration is proportional to the concentration of antibody of interest [58].

Analytical performances of apoB immunoassays methods are better than measurement or calculation of LDL-C and non-HDL-C [59]. In general, LDL-C, non-HDL-C, and apoB concentrations are highly correlated and provide similar information about ASCVD risk [60]. In patients with high TG, measurement of LDL-C underestimates the risk of ASCVD as TG-rich remnant particles are also apoB-particles, such as those with very low levels of LDLs. It will underestimate concentration of total cholesterol carried by LDLs and underestimate the total concentration of apoB-particles. Therefore, around 20% of patients have discordance between measured LDL-C and apoB assessment [61]. Considering this potential inaccuracy of LDL-C in dyslipidaemia, measurement of apoB and non-HDL-C is recommended for patients with diabetes, hypertriglyceridemia, or very low LDL-C. ApoB provides an accurate estimate of the total concentration of atherogenic particles under all circumstances.

Concerning current measurement of apoA-1, immunoassays seem to provide a sufficient estimation of HDL-C concentration and ratios between atherogenic lipoproteins (non-HDL-C, apoB) and HDL-C or apoA-1 are useful for risk estimation, but not for diagnosis or as treatment targets [45]. The relationship between the increase in HDL/apoA-1 levels and a potential antiatherogenic function of HDL particles is not fully understood. Even if observational studies have demonstrated that a high level of HDLs is associated with reduced risk of CVD, many attempts have been made to raise HDL levels as a therapeutic approach for patients with atherosclerosis, but these were not successful.

Reliable biomarkers for atherosclerotic diseases are available among the inflammatory and oxidative stress components of atherosclerosis, in combination with lipids in the arterial wall [62]. From a fundamental point of view, all elements resulting from or responsible for each stage of the disease are potentially interesting, from the endothelial dysfunction to the rupture of atheromatous plaque. These biomarkers involved in the development of atherosclerosis are causal factors of the disease. The use of these biomarkers can allow individual risk assessment and individual monitoring of treatment [63]—most of the population is concerned by general guidelines, but some situations cannot be solved this way. There are cases where non-traditional factors are needed to properly assess cardiovascular risk [63]. Specific biomarkers to atherosclerotic events such as MPO and oxidized-LDLs (ox-LDLs) can be then interesting [64].

Different types of immunoassays for ox-LDLs already exist, based on the ELISA kit [65]. These tests use mAb directed against different epitopes: aldehyde-modified LDL (4E6) [66] and LDL phosphatidylcholine (DLH3 and E06). DLH3 binds to oxidized epitopes of phosphatidylcholines that form complexes neighboring apoB protein [67]. E06 binds phosphatidylcholines containing oxidized phospholipids, next to apoB [68]. Results suggest that circulating ox-LDL, measured with antibody 4E6, is less predictive in development of CVD than apoB and cholesterol/HDL-C ratio [69]. Four mAb against human Mox-LDLs have already been produced. These antibodies have been used to study the correlation of plasma level Mox-LDLs with angiotensin II, adiponectin, and myeloperoxidase activity [70]. Lipid peroxidation mediated by MPO leads to formation of malondialdehyde (MDA)-modified LDLs and HDLs that can also be measured by ELISA kits [71,72]. An ELISA kit commercialized has been used to measure Mox-HDLs in its lipid oxidation form and is negatively correlated with antioxidant activity [73]. The concept of dysfunctional HDLs is calculated from the ratio of oxidized-HDLs to HDLs. A high affinity mAb has been developed specifically recognizing oxTrp 72 from Mox-HDLs [35].

### 2.2. Treatment Targets

Apolipoprotein analysis has the disadvantage of not being included in algorithms for calculation of global risk, and it has not been a predefined treatment target in controlled trials. However, with the past and recent results concerning apolipoproteins, and their modifications in case of atherosclerosis, the use of those markers must be considered to improve the assessment of cardiovascular risks. The literature suggests the key role of MPO-modified apolipoproteins in atherosclerotic plaque formation. It could be then interesting to investigate not only the quantity of lipoproteins but also their quality. Apolipoproteins can provide more information than lipoproteins, concerning the number of apoparticles and the functionality of lipoproteins [63]. Peng et al. have shown that oxidized-apoA-1 is highly correlated with cardiovascular events [74]. Then, it should be relevant to strengthen the assessment of lipoprotein quantity with knowledge about their quality. The evaluation of ratio of functional and native lipoproteins compared to unfunctional and oxidized ones can be a relevant measurement. There is a constant evolution of guidelines and healthcare system, and in the context of personalized medicine, an improvement of cardiovascular risks assessment should be carefully considered especially for prevention. However, implementing new biomarkers asks analytical and clinical validations that are worthwhile in the view of the encouraging recent studies.

Current methods to assess blood lipids are numerous and may seem appropriate now. With the advancement of new technologies, the emergence of new therapeutic goals and our knowledge of the role of oxidized apolipoproteins, it is necessary to update these assays to meet future demands.

## 3. Upgrading Current Methods with Mass Spectrometry

Since the National Heart, Lung and Blood Institute (NHLBI) launched a National Cholesterol Education Program (NCEP) in the United States in 1985, many cardiovascular risk prevention programs have been published by national competent authorities [75]. They provide guidelines to reduce the incidence of high cholesterol. This sensitization to screening has resulted in the growth of the market for cholesterol test kits for routine laboratory tests or to be used directly by patients [75].

### 3.1. Issues with Current Methods

Homogeneous assays represent the latest generation of fully automated methods using specific reagents to expose and measure lipoproteins. They separate HDLs and LDLs/VLDLs by using their biochemical and physical characteristics. Tests exist for routine uses in clinical laboratories and they generally meet NCEP criteria for accuracy, precision and total error [76]. Immunochemical methods with good analytical performance are available for apolipoprotein assessment and are easily run in clinical analyzers. In addition, these tests are not sensitive to high level of TG.

This generation of homogeneous assays, revolutionary in the historical context of the evolution of HDL-C assays, has been critically examined [75,77,78]. The constant need to control costs in clinical laboratories as well as the requirements for analytical performance have created a demand for more and more efficient methods. Indeed, discordant results compared to the reference methods have been observed with some tests, and the sources of the discrepancies are not well characterized. Deviations between different laboratories have also been observed [76]. Some reagents have not been fully evaluated or standardized and others are still being optimized. Therefore, homogeneous tests cannot be reliably recommended for use in long-term clinical trials and other research applications without in-depth validation.

LDL-C concentration is still assessed by using the Friedewald calculation in most clinical laboratories. This approach has several limitations and does not always meet the current total error recommendation in LDL-C measurement [79]. The classical homogeneous LDL-C assay is an alternative for measuring LDL-C. It allows the measurement of LDL-C in samples with hypertriglyceridemia or samples collected postprandially, with precision and acceptable accuracy [79].

Concentrations of apoA-1 and apoB-100 are currently measured by antibody-based immunoassay methods [80]. Immunoassays have a lot of disadvantages, such as the high development cost, kit formats and the indirect detection with antibodies recognizing specific epitopes on the interest proteins that provide many limitations [81]. There is a lack of concordance between sets of reagents that do not measure the same concentration. Epitopes on analyte can be blocked from reagent antibodies by endogenous immunoglobulins, such as autoantibodies. The latter can act in a similar way to anti-reagent antibodies by binding reagent antibodies. Reagent antibodies can also be saturated with analyte preventing sandwich formation [81]. Consequently, mass spectrometry (MS) has been proposed as a potential alternative to assess apolipoproteins, where peptides will be detected, instead of directly targeting apo with antibodies [82].

### 3.2. Comparison between Immunoassay and Mass Spectrometry for Protein Analysis

The choice between immunoassay and mass spectrometry methods depends on the needs, goals, and patient population of the laboratory. These assays are both originally used to detect proteins, hormones, and other small molecules. The selection of the most adequate method depends on the analyte and the advantages and disadvantages of each method. In general, most immunoassays have lower limits of quantification than MS assays and can easily reach detection limits in the picogram per milliliter in blood samples [83]. MS based methods need extensive sample clean up and low flow analyses to reach the same performance [84]. Both methods are subject to matrix effects, but this effect can be reduced with prior sample extraction. Immunoassays are more susceptible to interference and cost more to run. MS used to require a specialized training to be employed, new developments made the automatization possible for MS, such as immunoassays [85]. Concerning the quantitation of protein biomarkers, it has been demonstrated that MS has clear advantages compared to immunoassays [86]. It is highly customizable depending on the proteins of interest. Therefore, MS methods can be developed by laboratories and can slightly differ from one another. Immunoassay kits are commercially developed with reagents, antibodies and standards that can change depending on the manufacturers. MS is a suitable candidate alternative to replace or improve immunoassay methods. It could provide additional clinical information, with good analytical specificity and using low-cost consumables.

### 3.3. Development and Challenges of A Mass Spectrometry Method to Analyze Oxidized Proteins

LC-MS/MS approach offers the possibility to first separate the peptides of interest in the liquid chromatography system (LC) and to detect them by generating ions with a specific mass-to-charge ratio (*m*/*z*). The ionized peptide is consequently selected for fragmentation (MS/MS), providing specific peptide fragments and with this additional specificity, the quality of peptide identifications from complex protein mixtures. The quantitative detection of the peptide is based on the abundance of the ion peptide and its fragments detected by the mass spectrometer. Limitations of the technique exist regarding to quantification of proteins in complex biological matrices. The validation of this type of approach will require going through some bioanalytical challenges, but these can be overcome by optimizing LC and MS parameters. Moreover, the use as internal standards of stable isotope-labeled peptides provided by chemical synthesis ensures reliable results on the quantification of specific peptides of the targeted protein. The identification of the most specific and abundant MPO-modified peptides will allow to develop an adequate separation and acquisition method for, respectively, LC and MS. An optimized method of data acquisition can gather sensitivity, reproducibility and an optimal number of peptide identifications [82].

To propose an attractive MS method for clinical laboratories, it is essential to meet unmet clinical needs by proposing reliable analysis of appropriate candidate markers. As previously stated, precision medicine is growing, and current standard assays need to be upgraded concerning the assessment of cardiovascular disease risk. In parallel, fundamental research has demonstrated the relevant of investigating lipoprotein quality through MPO-modified apolipoproteins A-1 and B-100.

Specifications concerning analytical and clinical performance must be predefined to fit clinical purpose. Analytical performance parameters must be adapted to require clinical performance. In MS proteomic, data from clinical outcome studies are not yet available. Therefore, it has been proposed to use biological variation data from intraindividual and interindividual variations to establish goals for imprecision, bias, and total allowable error of the method. If the biological data sought are not available, analytical performance specifications should be set based on the “state-of-the-art” performance and the purpose and mode of the test in the clinical pathway [87]. Concerning LC-MS based protein quantitation, analytical coefficient of variation (CV) less than 10% can be possible [87]. EMA and FDA guidelines for validation of LC-MS/MS methods even allows 15% and 20% at the lower limit of quantification (LLOQ) [88].

Single- or multiplex quantitative approaches are well-known for protein assays [89]. If MS was first used for small molecules, it has been now well established as a protein quantification tool [90]. Its field of application is vast and varied, from monoclonal immunoglobulins detection to resolving quantification for example [86,87]. Multiplexing technologies such as LC-MS/MS allow the measurement of molecular marker patterns that confer significantly more information than the measurement of a single parameter alone [91]. That kind of MS analysis follows a bottom-up strategy, where peptide quantities are investigated to determine concentrations of protein targets for clinical interpretations. In contrast, a top-down approach is the analysis of intact proteins and their fragmentation. The bottom-up strategy involves protein digestion, and produced peptides are detected by MS/MS. During the method development, the selection of peptides is important as a fingerprint for the specificity and selectivity of the MS method. To guarantee accurate quantitation of proteins, a minimum of two peptides measured per protein is recommended at the end of the method development [92]. It is desirable to select peptides without missed cleavage variations or unstable amino acids such as methionine or tryptophan to ensure the accuracy of the method. In the context of post-translational oxidative modifications (oxPTM) such as MPO-modified apoA-1 and B-100, additional peptides are required to quantify oxPTM and monitor the quality of the apolipoproteins. Peptides containing methionine, tryptophane and tyrosine, potentially modified by MPO, must be selected. However, it brings an additional challenge concerning the stability of these peptides during collection, storage, and preparation of samples, as they are also subject to chemical modifications. The analysis of the post-translational modification is challenging but there is a constant evolution of methodologies facilitating the identification of oxPTM. Therefore, targeted mass spectrometry programs have been developed to identify ion-containing peptides, indicating the presence of oxidative modifications such as Cl-Tyr, oxTrp or oxMet [93]. The objective is to be able to quantify the level of oxPTM, either in absolute terms or relative compared to total level of proteins (semi-quantitative). This will allow the quality of the analyzed proteins to be assessed in the context of a pathophysiological situation.

Quantitation and confirmation of these peptides are ensured using two optimized multiple-reaction-monitoring (MRM) transitions (Table 3). The specific pair of mass-to-charge ratio (*m*/*z*) values associated with the precursor and product ions selected cannot be mistaken with another compound from the samples, to avoid interferences. The most abundant product ion is used as the quantifier for quantitation of the corresponding peptide. The second most abundant and a specific one is used as a qualifier to confirm the identification of the peptide. The quantitation of apolipoproteins require analysis of a control peptides, which do not carry amino acid subject to MPO-modifications and therefore only depend on quantity of the protein. Qualitative assessment is ensured using potentially in vivo MPO-modified peptides. MPO activity on apoA-1 and apoB-100 is evaluated by a ratio between peptides carrying MPO modifications and the corresponding peptides without alteration (Figure 2). A peptide is identified to be MPO-modified when its *m*/*z* ratio is superior to the *m*/*z* of its native form. The difference corresponds to the addition of oxygen during the oxidation. In addition, oxidized peptides are eluted from the column earlier than their unmodified form. For example, in Table 3, W72 Ox appears at 10.3 min, while its native form W72 is detected at 10.7 min. This retention time shift can also help to distinguish a difference between in vivo and artificial oxidation from sample preparation or analysis. Therefore, three kinds of peptides are examined to assess concomitantly quantity and quality of apoA-1 and B-100: control peptides, native peptides and oxidized peptides.

The use of samples such as serum or plasma enables alignment with clinical laboratories. Blood samples are challenging as the matrix can provide diagnostic errors that need to be prevented, such as interferences from blood compounds, alteration of enzymatic digestion and stability of proteins. Sample preparation must be optimized to solve potential interferences (Figure 3). Standard procedure for protein digestion implies denaturation and reduction in the proteins, followed by alkylation of free thiol groups [87]. When proteins are completely unfolded, they are easily accessible by the trypsin for homogeneous digestion at the carboxyl side of the amino acids lysine (K) or arginine (R). Tryptic activity is optimal at pH 8 and at a temperature of 37 °C. The presence of surfactant must be minimized by removing it or diluting the solution to be digested. The digestion time is important and is determined depending on the digestion time curve of peptides, with a balance between an optimum digestion and minimum degradation of peptides. The origin of the trypsin may impact the digestion and affects the quantity of trypsin used. Depending if the trypsin is from bovine pancreas or is sequencing-grade modified, the specificity and efficacy of trypsin may change and modifies the quantity if trypsin needed, that is generally set around a trypsin:protein ratio of 1:40 [94]. Even if trypsin is the most used enzyme in proteomics, other proteases can be used to generate proteotypical peptides such as chymotrypsin and Glu C. These enzymes have complementary cleavage specificity and generate different peptides with oxidized residue than trypsin. They can provide a good alternative to trypsin in special circumstances or can be added to the digestion mixture for synergetic action. Some peptides from apolipoproteins are subject to missed cleavages by trypsin, especially when sequence motifs have successive lysine and/or arginine residues [95]. For example, the residue of W108 from apoA-1 is localized in the sequence KKWQEEMELYRQ. The main peptide obtained after digestion with trypsin is WQEEMELYR, but the peptide KWQEEMELYR can also be produced. It is then necessary to be careful with the sequence of the targeted peptides.

Good analytical performance requires the use of internal standard, to correct losses of the target peptides. Internal standards are usually stable isotope labelled (SIL) compound with the same structure as the measured analyte. There are different strategies, from SIL peptide to SIL protein. The use of entire SIL protein is considered as a gold standard for the accurate quantitation of tryptic peptides by LC-MS/MS method, but it is not always available or is extremely costly [96]. The addition of SIL peptides is therefore the most common alternative and the ratio between endogenous peptides and SIL peptides is expected to be constant despite the digestion or the sample preparation workflow.

If the sample matrix brings too many interferences regarding the digestion, the separation on the analytical column or the detection with the MS, supplementary steps can be added before sample preparation or injection. These problems can be encountered while analyzing specific proteins from blood samples that are initially rich with soluble proteins such as albumin. Prior purification of the samples at the beginning of sample treatment (separation/isolation following the features of the target analyte) or after digestion (sample clean up based on solid phase extraction) can improve analytical performance or ensure the protection of the instruments. The removal of the major part of a complex matrix will reduce the interferences created by the noise. Noise produces fake peaks or can hide small intensities in the measurements [97].

Implementation of all LC-MS equipment requires important investment at first, regarding cost of equipment and its maintenance. It is important to establish quality protocols to ensure the system suitability. However, most of routine clinical laboratories already possess mass spectrometers and the addition of new LC-MS/MS methods is not problematic. MS-based method does not require the expensive development of new antibodies and remains independent of the quality of antibody batch production. Even if the purchase of such equipment is a significant investment, its implementation in clinical chemistry laboratories becomes possible. The great capacity to adapt of MS methods allows the analysis of a large range of proteins with small modifications. Amongst the prospects of extension to multiplex protein measurements, the establishment of patient profiles has to be considered [98]. The increasing interest in the development of MS-based proteomic methods offers a lot of perspectives for commonly used protein biomarkers and new ones [99].

### 3.4. Mass Spectrometry Allows Quantitation of Serum ApoA-1 and ApoB-100

Several LC-MS/MS methods have already been developed in order to assess the quantity of apolipoproteins in serum/plasma. They all demonstrate the high-throughput potential of LC-MS/MS to quantify serum proteins without antibodies. By comparing with immunoassays, it has been concluded that the MS method provides a reliable alternative for a clinical use for apolipoproteins quantification [100,101].

One of the most recent LC-MS/MS methods that have been developed in the context of clinical chemistry requirements for the accurate quantitation of serum apoA-1 and apoB-100 is by Smit and Cobbaerts [92]. After the review of current methods for assessing lipid status, they seized the opportunity to propose an antibody-independent measurement of protein quantity via specific signature peptides. Therefore, they presented the development of a Tier 1 application to use in clinical chemistry. As already detailed in Figure 2, samples are first prepared with trypsin digestion to enable the signatures peptides to be measurable. The metrological traceability of the tests is guaranteed with the use of value-assigned calibrators, prepared in the same way as patient samples. These samples are well characterized and are traceable to World Health Organization international reference materials. A linearity range have been primary established, from 0.1 nmol/L to 800 nmol/L for apoA-1, and from 0.0125 nmol/L to 40 nmol/L for apoB-100, in a normal pooled serum digest. The difference between these concentration ranges is explained by the difference in molecular weight between apoA-1 (around 29 kDa) and apoB-100 (around 550 kDa) [94,95]. As they both have a physiological mass concentration range approximatively between 1.0 and 1.6 g/L, there are fewer apoB-100 particles than apoA-1 [102]. Moreover, mixtures of two value-assigned calibrators (VAC) at five concentration levels have been used to check the linearity of peptide measurements after protein digestion. The chosen VACs were characterized by high and low concentrations of apoA-1 and apoB-100 and have been used to recover a large physiological range of concentrations. The highest and lowest concentrations of apoA-1 were, respectively, 1.15 and 1.60 g/L, and the concentrations of the 25/75, 50/50 and 75/25 (*v*/*v*) mixtures were 1.49, 1.38 and 1.26 g/L. For apoB-100, the concentrations were 0.76 and 1.29 g/L for the VACs, and 1.16, 1.03 and 0.89 g/L for the mixtures. Comparison between the target levels and the calculated ones has been made. Amongst all peptides tested for apoB-100 quantification, peptides FPEVD and TEVIP showed linearity with squared correlation coefficients (R^2^) > 0.994. Concerning apoA-1, AKPA and QGLLP had the best R^2^ (≥0.945). Minimal and maximal recovery for the apoA-1 peptides ranged between 96.7% and 103.9%, and between 97.8% and 102.6% for the apoB-100 peptides. Imprecision of apolipoproteins quantitation has been evaluated, with an overall imprecision for the apoA-1 peptides varying between 4.2% and 8.6%, and between 4.2% and 6.4% for the apoB-100 peptides. Finally, a comparison between the LC-MS/MS and immunoturbidimetric method showed equivalent results. Correlations were found with R^2^ > 0.94 (for apoA-1 peptides) and >0.96 (for apoB-100 peptides). The mean apoA-1 and apoB-100 concentrations, obtained with external calibration, were higher (from 1.9% to 7.5%) than the mean ITA values. The average biases met the minimal allowable bias criterion of 5.6% for apoA-1 and 9.0% for apo B100. Test results meet the total error average (Tea) limit with minimal Tea criteria of 13.7% and 17.4% for apo A-I and apo B100, respectively. These tests led the research team to choose the signature peptides that allow the most reliable quantitation of apoA-1 and apoB-100. 

Through the development of a LC-MS/MS method to quantify apoA-100 and apoB-100, the conception of a similar method targeting MPO-modified peptides is possible, following the same validation pathway. As previously stated, there will be challenges concerning analytical performances and stability of samples. For the assessment of apoA-1 quality, oxidized Trp72 and Tyr 192 are clinically relevant while for apoB-100, no serious candidate has been highlighted yet. The next tough step will be the clinical validation of the signature oxidized peptides, combined with the signature native peptides to provide a simultaneous quantitative and qualitative assessment of apolipoproteins. As the absolute quantification of oxidized peptides represents a major obstacle due to a lot of constraints, a relative expression of them compared to their native form is an interesting option. Standardization of apolipoprotein quality through absolute quantitation of their oxidized counterpart may not be possible for now, but the use of a ratio between oxidized and native peptides could improve the diagnostic and the individual follow-up of each patient in the context of cardiovascular diseases.

The LC-MS/MS method is promising to make an important contribution to CVD risk assessment, especially in patients with dyslipidaemia. Clinical studies are needed to evaluate the possible usefulness of this quantitative and qualitative lipoprotein assay and what it could bring compared to current assays.

## 4. Conclusions and Perspectives

The evolution of research in the identification of mechanisms involved in the development of atherosclerosis has highlighted the importance of lipoproteins. As Mox-HDLs are dysfunctional and Mox-LDLs are proinflammatory, the analysis of lipoproteins became more specifically a matter of their quality. It is therefore essential to emphasize the quantification of oxidized apolipoproteins by LC-MS/MS which allows efficient and cost-effective methods to be developed for clinical biology analyses.

Current developed methods concern only quantification of native apolipoproteins, but it is possible to extend them to the analysis of their oxidative states. First, it is necessary to decide a consensus on the most relevant modifications in the development of atherosclerosis. Trp 72 is the main peptide of interest from apoA-1, regarding MPO activity. For apoB-100, there is still a need for agreement with relevant peptides targeted by MPO and, amongst them, which one is the most sensitive. It is also required to obtain a consensus between specificity and sensitivity. As already said, it is necessary to develop an optimized acquisition method to obtain an optimal LC-MS/MS quantitative-method. It includes the integrations of the most optimal MS-settings such as retention times, *m*/*z* of peptides carrying or not these modifications, the fragmentation parameters. It is important to exclude oxidation from sample processing and the mass spectrometry method of analysis. To provide quantitative information, the use of standards such as isotope labelled control peptides should be considered. Studying the ratio between the peptides of native apolipoproteins and the corresponding oxidized forms might provide information of the oxidative status of apolipoprotein and consequently, their quality.

Monoclonal antibodies have been developed to determine the impact of oxidized lipoproteins on pathophysiology of cardiovascular diseases. For apoA-1, the epitope includes Trp72, while for apoB-100, the targeted sequence of monoclonal antibodies has not been yet elucidated. Mass spectrometry allows, with small modifications in the method, the detection of any desired peptides instead of producing new antibodies for each interesting residue from apolipoproteins. In addition, the targets are well described as several characteristics of each peptide are needed in order to identify them in MS methods.

Concerning analytical validation, a model of LC-MS/MS approach have been developed) to quantify apoA-1 and apoB-100 in serum. Signature peptides have been selected, and corresponding stable isotope labelled peptides have been used as internal standards. The measure of these peptides in serum digests shows good linearity in the physiologically relevant concentration ranges. Native value-assigned sera can be used as external calibrators for method verification. Correlation of LC-MS/MS results with immunoturbidimetric assay results, for normo- and hypertriglyceridemic samples, is significantly positive for both apoA-1 and apoB-100. There is still a lot of work to do concerning measurement of oxidized apolipoproteins, but it is likely that such a method may lead to improvements in the assessment of cardiovascular risk [92].

For the clinical significance, the method should be applied on population with and without cardiovascular risks. The impact of MPO-modified apolipoproteins will have to be described and categorized. The analysis of biological specimens used from cohorts studied in previous studies, such as the Framingham Heart Study, is a great opportunity to determine the place of apolipoprotein measurements. At this stage, the mean difficulty is to prove that oxidized apoA-1 and B-100 can significantly impact the cardiovascular risk assessment or at least reduce the residual risk. Indeed, a biomarker is evaluated as useful depending on its capacity to influence the prediction of risk in primary prevention. At the moment, adding apoB to a standard 10-year risk prediction model will not meaningfully change the statistic [63]. Current guidelines based on these algorithms are applicable to a major part of the population, but some cases need more investigations. In this context, measurements of apolipoprotein quantity and quality could provide better assessment of CV risks.

## Figures and Tables

**Figure 1 molecules-26-05264-f001:**
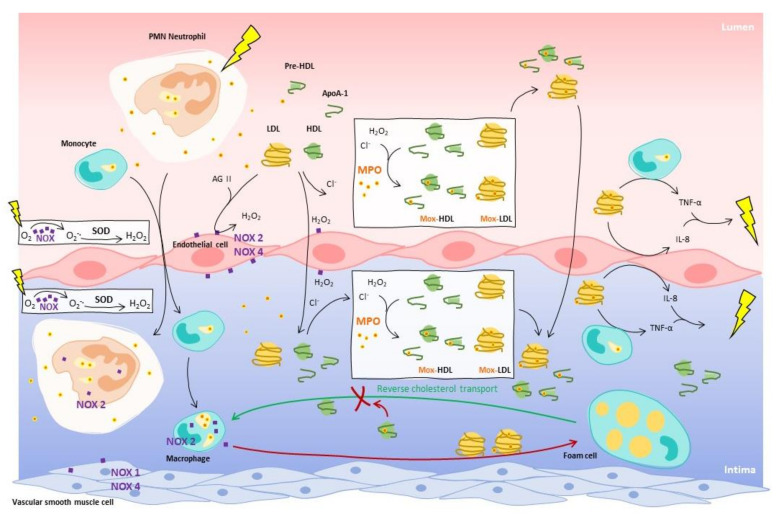
Activation of endothelium by LDLs promotes migration of monocytes and lymphocytes (neutrophils) into the underlying intima, and monocytes are then differentiated in macrophages. MPO can be found in neutrophils and macrophages. With the presence of Nox on endothelial cells, neutrophils and macrophages, H_2_O_2_ is produced by consuming the burst of oxygen from the inflammation. MPO interacts with LDLs and HDLs in both lumen and intima to oxidize apolipoproteins using chloride ions and H_2_O_2_ and produces Mox-LDLs and Mox-HDLs. While HDLs promote reverse cholesterol transport (RCT), Mox-HDLs are unable to perform the RCT and promote formation of foam cells, such as Mox-LDLs. Furthermore, Mox-LDLs induce pro-inflammatory activity from endothelial cells by stimulating the production of IL-8, and from monocytes by generation of TNF-α. MPO modifications contribute to maintain the inflammation in the vascular wall and in the vascular lumen.

**Figure 2 molecules-26-05264-f002:**
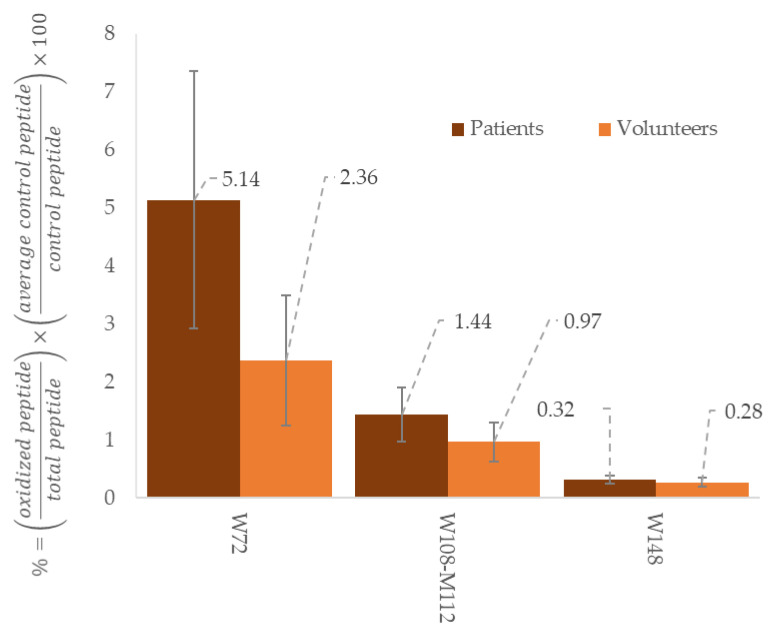
Comparison between healthy volunteers (*n* = 3) and patients in haemodialysis (*n* = 3) regarding to oxidation ratio of different peptides from apoA-1. These results come from preliminary assays made in our laboratory to investigate signature peptides relevant with cardiovascular risks. The results are expressed in ratio of oxidized peptides with total peptides (native and oxidized), adjusted with a control peptide and expressed in %.

**Figure 3 molecules-26-05264-f003:**
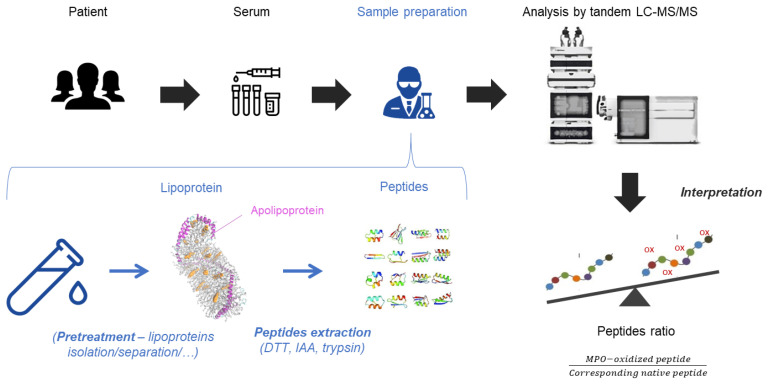
From serum to peptides analysis. Serum samples content lipoproteins with apolipoproteins on their surface. If needed, a pre-treatment to isolate/separate lipoproteins can be used. In order to extract peptides from apolipoproteins, a proteomic sample treatment is applied with dithiothreitol (DTT), iodoacetamide (IAA) and trypsin. The mix of peptides is analyzed by the tandem LC-MS/MS. The results are expressed in peptides ratio between the MPO-oxidized peptides and their corresponding native peptides.

**Table 1 molecules-26-05264-t001:** Summary of relevant modified residues on apoA-1 from the literature. We summarize in this table the main changes in apoA-I notified in the literature.

Modifications	Modified Residues	Tested Condition	Reference
Chlorophenylalanine	Phe 57, Phe 71	ApoA-1/HDL with MPO/H_2_O_2_/Cl^−^	[30]
Methionine sulfoxide	Met 86, Met 112, Met 148	ApoA-1/HDL with MPO/H_2_O_2_/Cl^−^	[30]
	Met 48, Met 112	ApoA-1 isolated from human atheroma tissue	[31]
Chlorotyrosine	Tyr 192, Tyr 236, Tyr 29, Tyr 18, Tyr 100, Tyr 115, Tyr 166	HDL with HOCl	[32]
	Tyr 192	HDL with MPO/H_2_O_2_/Cl^−^ or HOCl	[29]
	Tyr 192, Tyr 166	HDL with MPO/H_2_O_2_ (<50 μM)/Cl^−^	[33]
	Tyr 192, Tyr 166, Tyr 29, Tyr 236	HDL with MPO/H_2_O_2_ (<100 μM)/Cl^−^ or 100 μM HOCl	[33]
	Tyr 166	ApoA-1 in vivo	[33]
Nitrotyrosine	Tyr 192, Tyr 18, Tyr 29, Tyr 236, Tyr 100, Tyr 115, Tyr 166	ApoA-1 with MPO/H2O2/NO_2_^−^ or ONOO^−^	[29]
	Tyr 18, Tyr 29, Tyr 236, Tyr 100	HDL with MPO/H_2_O_2_/NO_2_^−^	[29]
	Tyr 192, Tyr 18, Tyr 29, Tyr 236, Tyr 115, Tyr 166	HDL with ONOO^−^	[29]
	Tyr 192, Tyr 166	HDL with MPO/H_2_O_2_ (<50 μM)/NO_2_^−^	[33]
	Tyr 192, Tyr 166, Tyr 29, Tyr 236	HDL with MPO/H_2_O_2_ (<100 μM)/NO_2_^−^	[33]
	Tyr 166, Tyr 18, Tyr 236	HDL with 100 μM peroxynitrite	[33]
	Tyr 192, Tyr 166	ApoA-1 in vivo	
Nitrotyrosine and methionine sulfoxide	Met 112 and Tyr 115 (single peptide)	ApoA-1 with MPO/H_2_O_2_/NO_2_^−^ or ONOO^−^	[29]
Chlorotyrosine and methionine sulfoxide	Tyr 192	ApoA-1 with MPO/H_2_O_2_/Cl^−^	[34]
Monohydroxytryptophan	Trp 8, Trp 50, Trp 72, Trp 108,	ApoA-1 isolated from human	[31]
	Trp 72		[35]
Dihydroxytryptophan	Trp 108		[31]
2-aminoadipic acid	Lysine		[31]

**Table 2 molecules-26-05264-t002:** Summary of relevant modified residues on apoB-100 from the literature. We summarize in this table the main changes in apoB-100 notified in the literature.

Modification	Tested Condition	Modified Residues	Reference
oxCys oxLys oxTrp oxMet	LDL oxidized by HOCl	Cys61/185/234/451/4190/3734/3890 Lys120 Trp1114/1210/1893/3567 Met3569	[39]
oxMet4	ApoB-100 in vitro by MPO	Met4	[40]
oxTy oxPhe	LDL in vivo	Tyr 103/413/666/2524/3490/3791/4088 Phe 3965	[41]
oxTy oxTrp	Hydroxyl radical and peroxynitrite	Tyr 583 and Trp 2524	[42]
oxTy	Hydroxyl radical and HOCl	Tyr 144, Tyr 276, Tyr 4451 and Tyr 4509	
oxTy	Hydroxyl radical and peroxynitrite and HOCl	Tyr 3295	
oxTyr oxTrp	HOCl	Tyr 3139 and Trp 3153	
oxTrp		Trp 4369	
oxMet oxTrp	Patients and volunteers	Met 4/4192 Trp 1114/3536	[43]
oxMet oxTrp Cl-Tyr dioxTrp	Patients only	Met 2499 Trp 2894/3606 Tyr 76/102/125/749 Trp 4369	
oxHis oxTrp oxLys	Patients	H2245, H2253, H3960 W1114 Lys293	[44]

**Table 3 molecules-26-05264-t003:** Transitions table for peptides from apoA-1 used to study MPO-modified apoA-1. Each line corresponds to a native or an oxidized (Ox) peptide carrying a residue of interest. The entire sequence of the peptide with the residue of interest, the retention time (RT), the *m*/*z* of precursor ion and the two product ions are defined. The site of fragmentation is detailed for each product ion, they are usually simple (+) or double (++) positively charged.

Peptide Sequence	Modification	RT (min)	Precursor Ion *m*/*z*	Product Ion 1	Product Ion 2
*m*/*z*	Frag	*m*/*z*	Frag
^46^LLDNWDSVTSTFSK^59^	W50	10.7	806.90	199.18	a2+	1271.59	y11+
^46^LLDNWDSVTSTFSK^59^	W50 Ox	9.93	814.89	199.18	1287.59
^62^EQLGPVTQEFWDNLEK^77^	W72	11	966.97	258.11	b2+	838.42	y14++
^62^EQLGPVTQEFWDNLEK^77^	W72 Ox	11.4	974.97	258.11	846.42
^108^WQEEMELYR^117^	W108 M112	9.2	642.29	969.43	y7+	315.15	b2+
^108^WQEEMELYR^117^	W108 M112 Ox	8.12	650.29	985.43	315.15
^108^WQEEMELYR^117^	W108 Ox M112	8.12	650.29	969.43	338.18	y2+
^140^LSPLGEEMR^149^	M148	8.5	516.26	416.20	y7++	621.26	y5+
^140^LSPLGEEMR^149^	M148 Ox	7.37	524.26	424.20	573.26	y5-64+
^216^QGLLPVLESFK^226^	Control 1	11.9	615.86	819.46	y7+		
^216^QGLLPVLESFK^226^	Control 1 labelled	12	619.36	826.48

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
