# Peer review of "Mass Spectrometry for the Monitoring of Lipoprotein Oxidations by Myeloperoxidase in Cardiovascular Diseases"

_molecules, 2021, doi:10.3390/molecules26175264_

Round 1
Reviewer 1 Report
Please see attached document

Author Response
Dear reviewer,
I have carefully read your comments and brought the expected modifications indicated in yellow. Unluckily, i cannot upload your file (with my answers) to associate it with the new version of the manuscript.
Thanks again for your help.
Catherine Coremans
Reviewer 2 Report
In my opinion the real review starts with paragraph 3, at page 9. Most of this article looks as a biochemical treaty on lipoproteins and atherosclerosis. While agreeing on the need for an introduction on this interesting topic, this introduction cannot represent 2/3 of the article. The authors should not forget that they have chosen as the title "MS for the monitoring....." and the only part of the text that reflects the title is from paragraph 3 onwards. I would have started this article with Figure 3 discussing all improvements that MS methods have introduced in the way of monitoring lipoprotein oxidations etc.
Unless the authors decide to make major changes to the text, the article, in its present form, cannot be accepted as a "MS monitoring of...."
Minor points:
-The last sentence of the Abstract is not correct. Being a review article, the authors should clarify whether, based on the articles present in the literature, MS can be considered a good tool in the field discussed or not.
-Lines 19-20 of the Abstract.This sentence is not clear. I understand that: "Once oxidized by MPO, modified lipoproteins lead to...." Is it correct ?
-Line 414. This sentence is not correct. I would suggest to delete "should be".
Author Response
Dear reviewer,
Thanks for your comments. We have been invited to write this review concerning our main research subject on MS-based method. It is a complicated topic to understand whitout having all the keys concerning the pathophysiology of the apolipoproteins in CV diseases, and why new approaches as MS are needed in clinical development. It seems interesting to us to make the link between basic research, clinical needs and analytical development. I nevertheless applied the modifications asked in green.
Thanks again for your time.
Round 2
Reviewer 2 Report
While not being completely convinced, I can accept the authors' justifications concerning the long introduction that I had challenged in my first revision.
The changes introduced in the text have improved the quality of the article.